# Nested Mini-Batch K-Means

**James Newling**
Idiap Research Institue & EPFL
james.newling@idiap.ch

**François Fleuret**
Idiap Research Institue & EPFL
francois.fleuret@idiap.ch

## Abstract

A new algorithm is proposed which accelerates the mini-batch k-means algorithm of Sculley (2010) by using the distance bounding approach of Elkan (2003). We argue that, when incorporating distance bounds into a mini-batch algorithm, already used data should preferentially be reused. To this end we propose using *nested* mini-batches, whereby data in a mini-batch at iteration $t$ is automatically reused at iteration $t + 1$.

Using nested mini-batches presents two difficulties. The first is that unbalanced use of data can bias estimates, which we resolve by ensuring that each data sample contributes exactly once to centroids. The second is in choosing mini-batch sizes, which we address by balancing premature fine-tuning of centroids with redundancy induced slow-down. Experiments show that the resulting `nmbatch` algorithm is very effective, often arriving within $1\%$ of the empirical minimum $100\times$ earlier than the standard mini-batch algorithm.

## 1 Introduction

The $k$-means problem is to find $k$ centroids to minimise the mean distance between samples and their nearest centroids. Specifically, given $N$ training samples $\mathcal{X} = \{x(1), \ldots, x(N)\}$ in vector space $\mathcal{V}$, one must find $\mathcal{C} = \{c(1), \ldots, c(k)\}$ in $\mathcal{V}$ to minimise energy $E$ defined by,

$$E(\mathcal{C}) = \frac{1}{N} \sum_{i=1}^{N} \|x(i) - c(a(i))\|^2, \tag{1}$$

where $a(i) = \arg\min_{j \in \{1,\ldots,k\}} \|x(i) - c(j)\|$. In general the $k$-means problem is NP-hard, and so a trade off must be made between low energy and low run time. The $k$-means problem arises in data compression, classification, density estimation, and many other areas.

A popular algorithm for $k$-means is Lloyd's algorithm, henceforth `lloyd`. It relies on a two-step iterative refinement technique. In the *assignment* step, each sample is assigned to the cluster whose centroid is nearest. In the *update* step, cluster centroids are updated in accordance with assigned samples. `lloyd` is also referred to as the *exact* algorithm, which can lead to confusion as it does not solve the $k$-means problem exactly. Similarly, *approximate* $k$-means algorithms often refer to algorithms which perform an approximation in either the assignment or the update step of `lloyd`.

### 1.1 Previous works on accelerating the exact algorithm

Several approaches for accelerating `lloyd` have been proposed, where the required computation is reduced without changing the final clustering. Hamerly (2010) shows that approaches relying on triangle inequality based distance bounds (Phillips, 2002; Elkan, 2003; Hamerly, 2010) always provide greater speed-ups than those based on spatial data structures (Pelleg and Moore, 1999; Kanungo et al., 2002). Improving bounding based methods remains an active area of research (Drake, 2013; Ding et al., 2015). We discuss the bounding based approach in § 2.1.

## 1.2 Previous approximate $k$-means algorithms

The assignment step of `lloyd` requires more computation than the update step. The majority of approximate algorithms thus focus on relaxing the assignment step, in one of two ways. The first is to assign all data approximately, so that centroids are updated using all data, but some samples may be incorrectly assigned. This is the approach used in Wang et al. (2012) with cluster closures. The second approach is to exactly assign a fraction of data at each iteration. This is the approach used in Agarwal et al. (2005), where a representative core-set is clustered, and in Bottou and Bengio (1995), and Sculley (2010), where random samples are drawn at each iteration. Using only a fraction of data is effective in reducing redundancy induced slow-downs.

The mini-batch $k$-means algorithm of Sculley (2010), henceforth `mbatch`, proceeds as follows. Centroids are initialised as a random selection of $k$ samples. Then at every iteration, $b$ of $N$ samples are selected uniformly at random and assigned to clusters. Cluster centroids are updated as the mean of all samples ever assigned to them, and are therefore running averages of assignments. Samples randomly selected more often have more influence on centroids as they reappear more frequently in running averages, although the law of large numbers smooths out any discrepancies in the long run. `mbatch` is presented in greater detail in § 2.2.

## 1.3 Our contribution

The underlying goal of this work is to accelerate `mbatch` by using triangle inequality based distance bounds. In so doing, we hope to merge the complementary strengths of two powerful and widely used approaches for accelerating `lloyd`.

The effective incorporation of bounds into `mbatch` requires a new sampling approach. To see this, first note that bounding can only accelerate the processing of samples which have already been visited, as the first visit is used to establish bounds. Next, note that the expected proportion of visits during the first epoch which are *re*visits is at most $1/e$, as shown in SM-A. Thus the majority of visits are first time visits and hence cannot be accelerated by bounds. However, for highly redundant datasets, `mbatch` often obtains satisfactory clustering in a single epoch, and so bounds need to be effective during the first epoch if they are to contribute more than a minor speed-up.

To better harness bounds, one must preferentially reuse already visited samples. To this end, we propose nested mini-batches. Specifically, letting $\mathcal{M}_t \subseteq \{1, \ldots, N\}$ be the mini-batch indices used at iteration $t \geq 1$, we enforce that $\mathcal{M}_t \subseteq \mathcal{M}_{t+1}$. One concern with nesting is that samples entering in early iterations have more influence than samples entering at late iterations, thereby introducing bias. To resolve this problem, we enforce that samples appear at most once in running averages. Specifically, when a sample is revisited, its old assignment is first removed before it is reassigned. The idea of nested mini-batches is discussed in § 3.1.

The second challenge introduced by using nested mini-batches is determining the size of $\mathcal{M}_t$. On the one hand, if $\mathcal{M}_t$ grows too slowly, then one may suffer from premature fine-tuning. Specifically, when updating centroids using $\mathcal{M}_t \subset \{1, \ldots, N\}$, one is using energy estimated on samples indexed by $\mathcal{M}_t$ as a proxy for energy over all $N$ training samples. If $\mathcal{M}_t$ is small and the energy estimate is poor, then minimising the energy estimate exactly is a waste of computation, as as soon as the mini-batch is augmented the proxy energy loss function will change. On the other hand, if $\mathcal{M}_t$ grows too rapidly, the problem of redundancy arises. Specifically, if centroid updates obtained with a small fraction of $\mathcal{M}_t$ are similar to the updates obtained with $\mathcal{M}_t$, then it is waste of computation using $\mathcal{M}_t$ in its entirety. These ideas are pursued in § 3.2.

## 2 Related works

### 2.1 Exact acceleration using the triangle inequality

The standard approach to perform the assignment step of `lloyd` requires $k$ distance calculations. The idea introduced in Elkan (2003) is to eliminate certain of these $k$ calculations by maintaining bounds on distances between samples and centroids. Several novel bounding based algorithms have since been proposed, the most recent being the `yinyang` algorithm of Ding et al. (2015). A thorough comparison of bounding based algorithms was presented in Drake (2013). We illustrate the basic

idea of Elkan (2003) in Alg. 1, where for every sample $i$, one maintains $k$ lower bounds, $l(i, j)$ for $j \in \{1, \ldots, k\}$, each bound satisfying $l(i, j) \leq \|x(i) - c(j)\|$. Before computing $\|x(i) - c(j)\|$ on line 4 of Alg. 1, one checks that $l(i, j) < d(i)$, where $d(i)$ is the distance from sample $i$ to the nearest currently found centroid. If $l(i, j) \geq d(i)$ then $\|x(i) - c(j)\| \geq d(i)$, and thus $j$ can automatically be eliminated as a nearest centroid candidate.

---

**Algorithm 1** `assignment-with-bounds`$(i)$

---

1: $d(i) \leftarrow \|x(i) - c(a(i))\|$          $\triangleright$ where $d(i)$ is distance to nearest centroid found so far
2: **for all** $j \in \{1, \ldots, k\} \setminus \{a(i)\}$ **do**
3:     **if** $l(i, j) < d(i)$ **then**
4:        $l(i, j) \leftarrow \|x(i) - c(j)\|$     $\triangleright$ make lower bound on distance between $x(i)$ and $c(j)$ tight
5:        **if** $l(i, j) < d(i)$ **then**
6:           $a(i) = j$
7:           $d(i) = l(i, j)$
8:        **end if**
9:     **end if**
10: **end for**

---

The fully-fledged algorithm of Elkan (2003) uses additional tests to the one shown in Alg. 1, and includes upper bounds and inter-centroid distances. The most recently published bounding based algorithm, `yinyang` of Ding et al. (2015), is like that of Elkan (2003) but does not maintain bounds on all $k$ distances to centroids, rather it maintains lower bounds on groups of centroids simultaneously.

To maintain the validity of bounds, after each centroid update one performs $l(i, j) \leftarrow l(i, j) - p(j)$, where $p(j)$ is the distance moved by centroid $j$ during the centroid update, the validity of this correction follows from the triangle inequality. Lower bounds are initialised as exact distances in the first iteration, and only in subsequent iterations can bounds help in eliminating distance calculations. Therefore, the algorithm of Elkan (2003) and its derivatives are all at least as slow as `lloyd` during the first iteration.

## 2.2 Mini-batch k-means

The work of Sculley (2010) introduces `mbatch`, presented in Alg. 4, as a scalable alternative to `lloyd`. Reusing notation, we let the mini-batch size be $b$, and the total number of assignments ever made to cluster $j$ be $v(j)$. Let $S(j)$ be the cumulative sum of data samples assigned to cluster $j$. The centroid update, line 9 of Alg. 4, is then $c(j) \leftarrow S(j)/v(j)$. Sculley (2010) present `mbatch` in the context sparse datasets, and at the end of each round an $l_1$-sparsification operation is performed to encourage sparsity. In this paper we are interested in `mbatch` in a more general context and do not consider sparsification.

---

**Algorithm 2** `initialise-c-S-v`

---

   **for** $j \in \{1, \ldots, k\}$ **do**
      $c(j) \leftarrow x(i)$ for some $i \in \{1, \ldots, N\}$
      $S(j) \leftarrow x(i)$
      $v(j) \leftarrow 1$
   **end for**

---

**Algorithm 3** `accumulate`$(i)$

---

   $S(a(i)) \leftarrow S(a(i)) + x(i)$
   $v(a(i)) \leftarrow v(a(i)) + 1$

---

# 3 Nested mini-batch k-means : `nmbatch`

The bottleneck of `mbatch` is the assignment step, on line 5 of Alg. 4, which requires $k$ distance calculations per sample. The underlying motivation of this paper is to reduce the number of distance calculations at assignment by using distance bounds. However, as already discussed in § 1.3, simply wrapping line 5 in a bound test would not result in much gain, as only a minority of visited samples would benefit from bounds in the first epoch. For this reason, we will replace random mini-batches at line 3 of Alg. 4 by nested mini-batches. This modification motivates a change to the running average centroid updates, discussed in Section 3.1. It also introduces the need for a scheme to

---
**Algorithm 4** `mbatch`

---
```
 1: initialise-c-S-v()
 2: while convergence criterion not satisfied do
 3:     M ← uniform random sample of size b from {1, ..., N}
 4:     for all i ∈ M do
 5:         a(i) ← arg min_{j∈{1,...,k}} ‖x(i) − c(j)‖
 6:         accumulate(i)
 7:     end for
 8:     for all j ∈ {1, ..., k} do
 9:         c(j) ← S(j)/v(j)
10:     end for
11: end while
```
---

choose mini-batch sizes, discussed in 3.2. The resulting algorithm, which we refer to as `nmbatch`, is presented in Alg. 5.

There is no random sampling in `nmbatch`, although an initial random shuffling of samples can be performed to remove any ordering that may exist. Let $b_t$ be the size of the mini-batch at iteration $t$, that is $b_t = |\mathcal{M}_t|$. We simply take $\mathcal{M}_t$ to be the first $b_t$ indices, that is $\mathcal{M}_t = \{1, \ldots, b_t\}$. Thus $\mathcal{M}_t \subseteq \mathcal{M}_{t+1}$ corresponds to $b_t \leq b_{t+1}$. Let $T$ be the number of iterations of `nmbatch` before terminating. We use as stopping criterion that no assignments change on the full training set, although this is not important and can be modified.

### 3.1 One sample, one vote : modifying cumulative sums to prevent duplicity

In `mbatch`, a sample used $n$ times makes $n$ contributions to one or more centroids, through line 6 of Alg. 4. Due to the extreme and systematic difference in the number of times samples are used with nested mini-batches, it is necessary to curtail any potential bias that duplicitous contribution may incur. To this end, we only alow a sample's most recent assignment to contribute to centroids. This is done by removing old assignments before samples are reused, shown on lines 15 and 16 of Alg. 5.

### 3.2 Finding the sweet spot : balancing premature fine-tuning with redundancy

We now discuss how to sensibly select mini-batch size $b_t$, where recall that the sample indices of the mini-batch at iteration $t$ are $\mathcal{M}_t = \{1, \ldots, b_t\}$. The only constraint imposed so far is that $b_t \leq b_{t+1}$ for $t \in \{1, \ldots, T-1\}$, that is that $b_t$ does not decrease. We consider two extreme schemes to illustrate the importance of finding a scheme where $b_t$ grows neither too rapidly nor too slowly.

The first extreme scheme is $b_t = N$ for $t \in \{1, \ldots, T\}$. This is just a return to full batch $k$-means, and thus redundancy is a problem, particularly at early iterations. The second extreme scheme, where $\mathcal{M}_t$ grows very slowly, is the following: if any assignment changes at iteration $t$, then $b_{t+1} = b_t$, otherwise $b_{t+1} = b_t + 1$. The problem with this second scheme is that computation may be wasted in finding centroids which accurately minimise the energy estimated on unrepresentative subsets of the full training set. This is what we refer to as premature fine-tuning.

To develop a scheme which balances redundancy and premature fine-tuning, we need to find sensible definitions for these terms. A first attempt might be to define them in terms of energy (1), as this is ultimately what we wish to minimise. Redundancy would correspond to a slow decrease in energy caused by long iteration times, and premature fine-tuning would correspond to approaching a local minimum of a poor proxy for (1). A difficulty with an energy based approach is that we do not want to compute (1) at each iteration and there is no clear way to quantify the underestimation of (1) using a mini-batch. We instead consider definitions based on centroid statistics.

### 3.2.1 Balancing intra-cluster standard deviation with centroid displacement

Let $c_t(j)$ denote centroid $j$ at iteration $t$, and let $c_{t+1}(j|b)$ be $c_{t+1}(j)$ when $\mathcal{M}_{t+1} = \{1, \ldots, b\}$, so that $c_{t+1}(j|b)$ is the update to $c_t(j)$ using samples $\{x(1), \ldots, x(b)\}$. Consider two options,

**Algorithm 5** `nmbatch`

---
1: $t = 1$            ▷ Iteration number
2: $\mathcal{M}_0 \leftarrow \{\}$
3: $\mathcal{M}_1 \leftarrow \{1, \ldots, b_s\}$            ▷ Indices of samples in current mini-batch
4: `initialise-c-S-v()`
5: **for** $j \in \{1, \ldots, k\}$ **do**
6:     $sse(j) \leftarrow 0$            ▷ Initialise sum of squares of samples in cluster $j$
7: **end for**
8: **while** stop condition is false **do**
9:     **for** $i \in \mathcal{M}_{t-1}$ and $j \in \{1, \ldots, k\}$ **do**
10:        $l(i, j) \leftarrow l(i, j) - p(j)$            ▷ Update bounds of reused samples
11:     **end for**
12:     **for** $i \in \mathcal{M}_{t-1}$ **do**
13:        $a_{old}(i) \leftarrow a(i)$
14:        $sse(a_{old}(i)) \leftarrow sse(a_{old}(i)) - d(i)^2$     ▷ Remove expired $sse$, $S$ and $v$ contributions
15:        $S(a_{old}(i)) \leftarrow S(a_{old}(i)) - x(i)$
16:        $v(a_{old}(i)) \leftarrow v(a_{old}(i)) - 1$
17:        `assignment-with-bounds(i)`            ▷ Reset assignment $a(i)$
18:        `accumulate(i)`
19:        $sse(a(i)) \leftarrow sse(a(i)) + d(i)^2$
20:     **end for**
21:     **for** $i \in \mathcal{M}_t \setminus \mathcal{M}_{t-1}$ and $j \in \{1, \ldots, k\}$ **do**
22:        $l(i, j) \leftarrow \|x(i) - c(j)\|$            ▷ Tight initialisation for new samples
23:     **end for**
24:     **for** $i \in \mathcal{M}_t \setminus \mathcal{M}_{t-1}$ **do**
25:        $a(i) \leftarrow \arg\min_{j \in \{1, \ldots, k\}} l(i, j)$
26:        $d(i) \leftarrow l(i, a(i))$
27:        `accumulate(i)`
28:        $sse(a(i)) \leftarrow sse(a(i)) + d(i)^2$
29:     **end for**
30:     **for** $j \in \{1, \ldots, k\}$ **do**
31:        $\hat{\sigma}_C(j) \leftarrow \sqrt{(sse(j))/(v(j)(v(j) - 1))}$
32:        $c_{old}(j) \leftarrow c(j)$
33:        $c(j) \leftarrow S(j)/v(j)$
34:        $p(j) \leftarrow \|c(j) - c_{old}(j)\|$
35:     **end for**
36:     **if** $\min_{j \in \{1, \ldots, k\}} (\hat{\sigma}_c(j)/p(j)) > \rho$ **then**            ▷ Check doubling condition
37:        $\mathcal{M}_{t+1} \leftarrow \{1, \ldots, \min(2|\mathcal{M}_t|, N)\}$
38:     **else**
39:        $\mathcal{M}_{t+1} \leftarrow \mathcal{M}_t$
40:     **end if**
41:     $t \leftarrow t + 1$
42: **end while**

---

$b_{t+1} = b_t$ with resulting update $c_{t+1}(j|b_t)$, and $b_{t+1} = 2b_t$ with update $c_{t+1}(j|2b_t)$. If,

$$\|c_{t+1}(j|2b_t) - c_{t+1}(j|b_t)\| \ll \|c_t(j) - c_{t+1}(j|b_t)\|, \tag{2}$$

then it makes little difference if centroid $j$ is updated with $b_{t+1} = b_t$ or $b_{t+1} = 2b_t$, as illustrated in Figure 1, left. Using $b_{t+1} = 2b_t$ would therefore be redundant. If on the other hand,

$$\|c_{t+1}(j|2b_t) - c_{t+1}(j|b_t)\| \gg \|c_t(j) - c_{t+1}(j|b_t)\|, \tag{3}$$

this suggests premature fine-tuning, as illustrated in Figure 1, right. Balancing redundancy and premature fine-tuning thus equates to balancing the terms on the left and right hand sides of (2) and (3). Let us denote by $\mathcal{M}_t(j)$ the indices of samples in $\mathcal{M}_t$ assigned to cluster $j$. In SM-B we show that the term on the left hand side of (2) and (3) can be estimated by $\frac{1}{2}\hat{\sigma}_C(j)$, where

$$\hat{\sigma}_C^2(j) = \frac{1}{|\mathcal{M}_t(j)|^2} \sum_{i \in \mathcal{M}_t(j)} \|x(i) - c_t(j)\|^2. \tag{4}$$

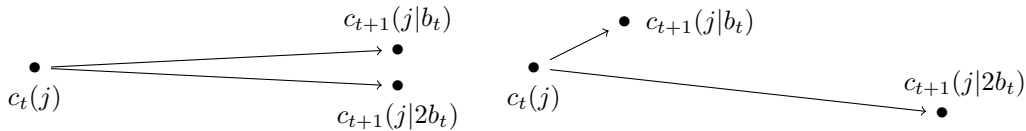

Figure 1: Centroid based definitions of redundancy and premature fine-tuning. Starting from centroid $c_t(j)$, the update can be performed with a mini-batch of size $b_t$ or $2b_t$. On the left, it makes little difference and so using all $2b_t$ points would be redundant. On the right, using $2b_t$ samples results in a much larger change to the centroid, suggesting that $c_t(j)$ is near to a local minimum of energy computed on $b_t$ points, corresponding to premature fine-tuning.

$\hat{\sigma}_C(j)$ may underestimate $\|c_{t+1}(j|2b_t) - c_{t+1}(j|b_t)\|$ as samples $\{x(b_{t+1}), \ldots, x(2b_t)\}$ have not been used by centroids at iteration $t$, however our goal here is to establish dimensional homogeneity. The right hand sides of (2) and (3) can be estimated by the distance moved by centroid $j$ in the preceding iteration, which we denote by $p(j)$. Balancing redundancy and premature fine-tuning thus equates to preventing $\hat{\sigma}_C(j)/p(j)$ from getting too large or too small.

It may be that $\hat{\sigma}_C(j)/p(j)$ differs significantly between clusters $j$. It is not possible to independently control the number of samples per cluster, and so a joint decision needs to be made by clusters as to whether or not to increase $b_t$. We choose to make the decision based on the minimum ratio, on line 37 of Alg. 5, as premature fine-tuning is less costly when performed on a small mini-batch, and so it makes sense to allow slowly converging centroids to catch-up with rapidly converging ones.

The decision to use a double-or-nothing scheme for growing the mini-batch is motivated by the fact that $\hat{\sigma}_C(j)$ drops by a constant factor when the mini-batch doubles in size. A linearly increasing mini-batch would be prone to premature fine-tuning as the mini-batch would not be able to grow rapidly enough.

Starting with an initial mini-batch size $b_0$, `nmbatch` iterates until $\min_j \hat{\sigma}_C(j)/p(j)$ is above some threshold $\rho$, at which point mini-batch size increases as $b_t \leftarrow \min(2b_t, N)$, shown on line 37 of Alg. 5. The mini-batch size is guaranteed to eventually reach $N$, as $p(j)$ eventually goes to zero. The doubling threshold $\rho$ reflects the relative costs of premature fine-tuning and redundancy.

### 3.3 A note on parallelisation

The parallelisation of `nmbatch` can be done in the same way as in `mbatch`, whereby a mini-batch is simply split into sub-mini-batches to be distributed. For `mbatch`, the only constraint on sub-mini-batches is that they are of equal size to guarantee equal processing times. With `nmbatch` the constraint is slightly stricter, as the time required to process a sample depends on its time of entry into the mini-batch, due to bounds. Samples from all iterations should thus be balanced, the constraint becoming that each sub-mini-batch contains an equal number of samples from $\mathcal{M}_t \setminus \mathcal{M}_{t-1}$ for all $t$.

## 4 Results

We have performed experiments on 3 dense datasets and sparse dataset used in Sculley (2010). The INFMNIST dataset (Loosli et al., 2007) is an extension of MNIST, consisting of $28 \times 28$ hand-written digits ($d = 784$). We use 400,000 such digits for performing $k$-means and 40,000 for computing a validation energy $E_V$. STL10P (Coates et al., 2011) consists of $6 \times 6 \times 3$ image patches ($d = 108$), we train with 960,000 patches and use 40,000 for validation. KDDC98 contains 75,000 training samples and 20,000 validation samples, in 310 dimensions. Finally, the sparse RCV1 dataset of Lewis et al. (2004) consists of data in 47,237 dimensions, with two partitions containing 781,265 and 23,149 samples respectively. As done in Sculley (2010), we use the larger partition to learn clusters.

The experimental setup used on each of the datasets is the following: for 20 random seeds, the training dataset is shuffled and the first $k$ datapoints are taken as initialising centroids. Then, for each of the algorithms, $k$-means is run on the shuffled training set. At regular intervals, a validation energy $E_V$ is computed on the validation set. The time taken to compute $E_V$ is not included in run times. The batchsize for `mbatch` and initial batchsize for `nmbatch` are $5,000$, and $k = 50$ clusters.

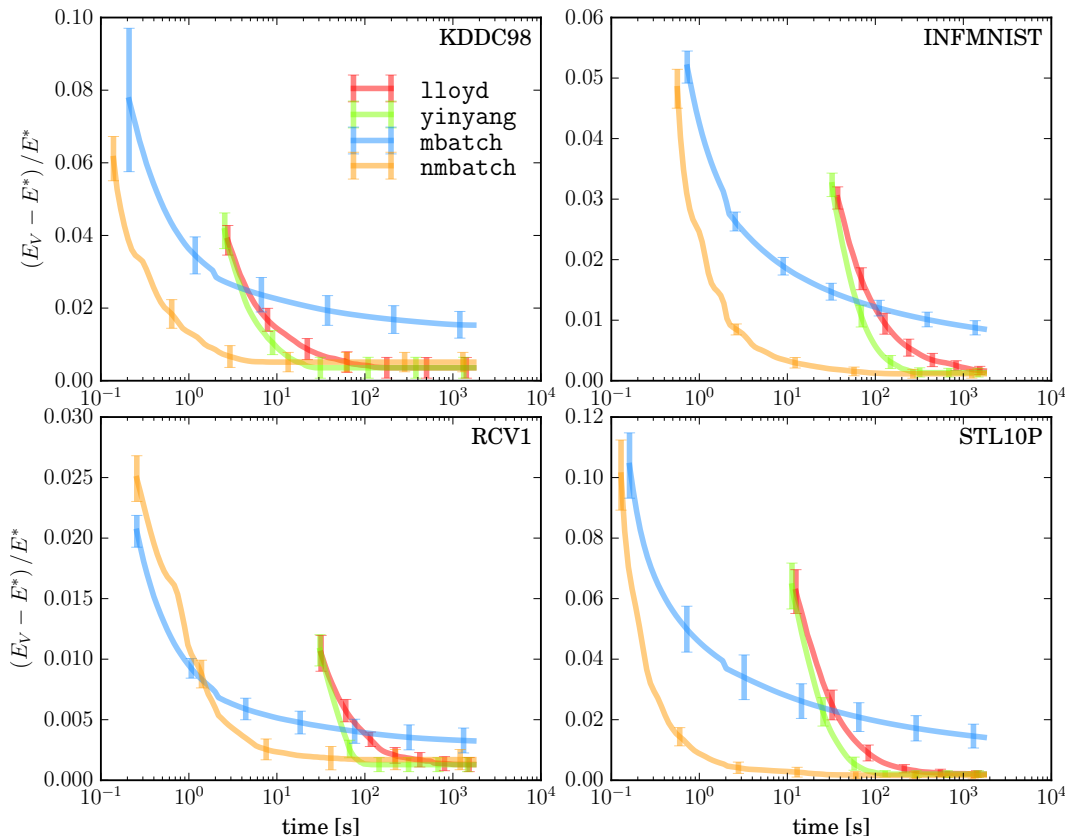

Figure 2: The mean energy on validation data ($E_V$) relative to lowest energy ($E^*$) across 20 runs with standard deviations. Baselines are `lloyd`, `yinyang`, and `mbatch`, shown with the new algorithm `nmbatch` with $\rho = 100$. We see that `nmbatch` is consistently faster than all baselines, and obtains final minima very similar to those obtained by the exact algorithms. On the sparse dataset RCV1, the speed-up is noticeable within $0.5\%$ of the empirical minimum $E^*$. On the three dense datasets, the speed-up over `mbatch` is between $10\times$ and $100\times$ at $2\%$ of $E^*$, with even greater speed-ups below $2\%$ where `nmbatch` converges very quickly to local minima.

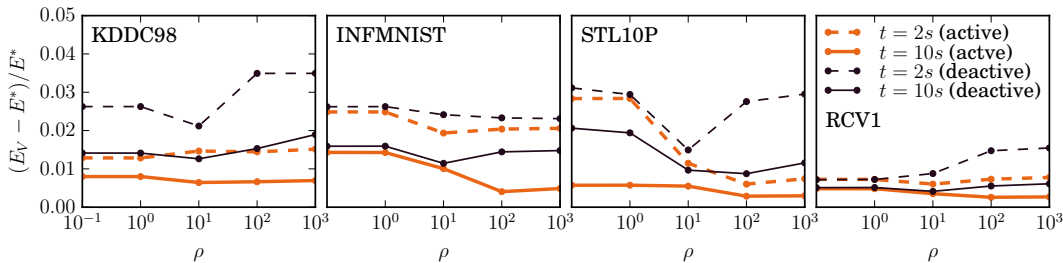

Figure 3: Relative errors on validation data at $t \in \{2, 10\}$, for `nmbatch` with and with bound tests, for $\rho \in \{10^{-1}, 10^0, 10^1, 10^2, 10^3\}$. In the standard case of active bound testing, large values of $\rho$ work well, as premature fine-tuning is less of a concern. However with the bound test deactivated, premature fine-tuning becomes costly for large $\rho$, and an optimal $\rho$ value is one which trades off redundancy ($\rho$ too small) and premature fine-tuning ($\rho$ too large).

The mean and standard deviation of $E_V$ over the 20 runs are computed, and this is what is plotted in Figure 2, relative to the lowest obtained validation energy over all runs on a dataset, $E^*$. Before comparing algorithms, we note that our implementation of the baseline `mbatch` is competitive with existing implementations, as shown in Appendix A.

In Figure 2, we plot time-energy curves for `nmbatch` with three baselines. We use $\rho = 100$, as described in the following paragraph. On the 3 dense datasets, we see that `nmbatch` is much faster than `mbatch`, obtaining a solution within 2% of $E^*$ between $10\times$ and $100\times$ earlier than `mbatch`. On the sparse dataset RCV1, the speed-up becomes noticeable within 0.5% of $E^*$. Note that in a single epoch `nmbatch` gets very near to $E^*$, whereas the full batch algorithms `lloyd` and `yinyang` only complete one iteration. The mean final energies of `nmbatch` and the exact algorithms are consistently within one initialisation standard deviation. This means that the random initialisation seed has a larger impact on final energy than the choose between `nmbatch` and an exact algorithm.

We now discuss the choice of $\rho$. Recall that the mini-batch size doubles when $\min_j \hat{\sigma}_C(j)/p(j) > \rho$. Thus a large $\rho$ means smaller $p(j)$s are needed to invoke a doubling, which means less robustness against premature fine-tuning. The relative costs of premature fine-tuning and redundancy are influenced by the use of bounds. Consider the case of premature fine-tuning with bounds. $p(j)$ becomes small, and thus bound tests become more effective as they decrease more slowly (line 10 of Alg. 5). Thus, while premature fine-tuning does result in more samples being visited than necessary, each visit is processed rapidly and so is less costly. We have found that taking $\rho$ to be large works well for `nmbatch`. Indeed, there is little difference in performance for $\rho \in \{10, 100, 1000\}$. To test that our formulation is sensible, we performed tests with the bound test (line 3 of Alg. 1) deactivated. When deactivated, $\rho = 10$ is in general better than larger values of $\rho$, as seen in Figure 3. Full time-energy curves for different $\rho$ values are provided in SM-C.

## 5 Conclusion and future work

We have shown how triangle inequality based bounding can be used to accelerate mini-batch $k$-means. The key is the use of nested batches, which enables rapid processing of already used samples. The idea of replacing uniformly sampled mini-batches with nested mini-batches is quite general, and applicable to other mini-batch algorithms. In particular, we believe that the sparse dictionary learning algorithm of Mairal et al. (2009) could benefit from nesting. One could also consider adapting nested mini-batches to stochastic gradient descent, although this is more speculative.

Celebi et al. (2013) show that specialised initialisation schemes such as k-means++ can result in better clusterings. While this is not the case for the datasets we have used, it would be interesting to consider adapting such initialisation schemes to the mini-batch context.

Our nested mini-batch algorithm `nmbatch` uses a very simple bounding scheme. We believe that further improvements could be obtained through more advanced bounding, and that the memory footprint of $O(KN)$ could be reduced by using a scheme where, as the mini-batch grows, the number of bounds maintained decreases, so that bounds on groups of clusters merge.

## A Comparing Baseline Implementations

We compare our implementation of `mbatch` with two publicly available implementations, that accompanying Sculley (2010) in C++, and that in scikit-learn Pedregosa et al. (2011), written in Cython. Comparisons are presented in Table 1, where our implementations are seen to be competitive. Experiments were all single threaded. Our C++ and Python code is available at `https://github.com/idiap/eakmeans`.

| INFMNIST (dense) | | RCV1 (sparse) | | |
|---|---|---|---|---|
| ours | sklearn | ours | sklearn | sofia |
| 12.4 | 20.6 | 15.2 | 63.6 | 23.3 |

Table 1: Comparing implementations of `mbatch` on INFMNIST (left) and RCV1 (right). Time in seconds to process $N$ datapoints, where $N = 400,000$ for INFMNIST and $N = 781,265$ for RCV1. Implementations are our own (ours), that in scikit-learn (sklearn), and that of Sculley (2010) (sofia).

## Acknowledgments

James Newling was funded by the Hasler Foundation under the grant 13018 MASH2.

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
