[Supplementary Material]

## SM-A  Showing that there are more first time visits than revisits in the first epoch

Let the probability that a sample is not visited in an epoch be $p$, where recall that an epoch consists of drawing $N/b$ mini-batches, where we assume $N \bmod b = 0$. Denote by $q$ the probability that the visit of a sample is a revisit. We argue that $q = p$ : the number of samples not visited exactly corresponds to the number of revisits, as the number of visits is the number of samples, by definition of an epoch. Clearly, $p = (1 - b/N)^{N/b}$, from which it can be shown that $1/4 \le p < 1/e$. Thus $q \le 1/e$ as we want. In other words, there are at least 1.718 first time visits for 1 revisit.

## SM-B  Showing that the expectations $\|c_{t+1}(j|2b_t) - c_{t+1}(j|b_t)\|^2$ and $\frac{1}{2}\hat{\sigma}_C^2$ are approximately the same

Recall that $\mathcal{M}_t(j)$ are the samples used to obtain $c_j(t)$, that is

$$c_t(j) = \frac{1}{|\mathcal{M}_t(j)|} \sum_{i \in \mathcal{M}_t(j)} x(i).$$

The mean squared distance of samples in $\mathcal{M}_t(j)$ to $c_j(t)$ we denote by $\hat{\sigma}_S^2(j)$,

$$\hat{\sigma}_S^2(j) = \frac{1}{|\mathcal{M}_t(j)|} \sum_{i \in \mathcal{M}_t(j)} \|x(i) - c_t(t)\|^2.$$

We compute the expectation of $\|c_{t+1}(j|2b_t) - c_{t+1}(j|b_t)\|^2$, where the expectation is over all possible shufflings of the data. Recall that $c_{t+1}(j|2b_t)$ is centroid $j$ at iteration $t+1$ if the mini-batch at size $t+1$ is $2b_t$, where $b_t$ is the mini-batch size at iteration $t$. Recall that we use $\mathcal{M}_t(j)$ to denote samples assigned to $c_t(j)$. We will now denote by $\mathcal{M}_{t+1}^{2b_t}(j)$ the sample indices assigned to $c_{t+1}(j|2b_t)$ and $\mathcal{M}_{t+1}^{b_t}(j)$ the sample indices assigned to $c_{t+1}(j|b_t)$. Thus,

$$\mathbb{E}\left(\ \|c_{t+1}(j|2b_t) - c_{t+1}(j|b_t)\|^2\right) =$$

$$= \mathbb{E}\left(\left\|\frac{1}{|\mathcal{M}_{t+1}^{2b_t}(j)|} \sum_{i \in \mathcal{M}_{t+1}^{2b_t}(j)} x(i) - \frac{1}{|\mathcal{M}_{t+1}^{b_t}(j)|} \sum_{i \in \mathcal{M}_{t+1}^{b_t}(j)} x(i)\right\|^2\right)$$

$$= \mathbb{E}\left(\left\|\frac{1}{|\mathcal{M}_{t+1}^{2b_t}(j)|} \sum_{i \in \mathcal{M}_{t+1}^{2b_t}(j)\setminus\mathcal{M}_{t+1}^{b_t}(j)} x(i) - \right.\right.$$

$$\left.\left.\left(\frac{1}{|\mathcal{M}_{t+1}^{b_t}(j)|} - \frac{1}{|\mathcal{M}_{t+1}^{2b_t}(j)|}\right) \sum_{i \in \mathcal{M}_{t+1}^{b_t}(j)} x(i)\right\|^2\right)$$

We now assume that the number of samples per centroid does not change significantly between iterations $t$ and $t+1$ for a fixed batch size, so that $|\mathcal{M}_{t+1}^{b_t}(j)| \approx |M_t(j)|$ and $|\mathcal{M}_{t+1}^{2b_t}(j)| \approx 2|M_t(j)|$. Continuing we have,

$$\approx \frac{1}{4|\mathcal{M}_t(j)|^2} \mathbb{E}\left( \left\| \sum_{i\in\mathcal{M}_{t+1}^{2b_t}(j)\backslash\mathcal{M}_{t+1}^{b_t}(j)} x(i) - \sum_{i\in\mathcal{M}_{t+1}^{b_t}(j)} x(i) \right\|^2 \right)$$

$$\approx \frac{1}{4|\mathcal{M}_t(j)|^2} \mathbb{E}\left( \left\| \sum_{i\in\mathcal{M}_{t+1}^{2b_t}(j)\backslash\mathcal{M}_{t+1}^{b_t}(j)} (x(i) - c_t(j)) \right. \right. -$$

$$\left. \left. \sum_{i\in\mathcal{M}_{t+1}^{b_t}(j)} (x(i) - c_t(j)) \right\|^2 \right)$$

The two summation terms are independant and the second has expectation approximately zero assuming the centroids do not move too much between rounds, so

$$\approx \frac{1}{4|\mathcal{M}_t(j)|} \left( \mathbb{E}\left( \frac{1}{|\mathcal{M}_t(j)|} \left\| \sum_{i\in\mathcal{M}_{t+1}^{2b_t}(j)\backslash\mathcal{M}_{t+1}^{b_t}(j)} (x(i) - c_t(j)) \right\|^2 \right) + \right.$$

$$\left. \mathbb{E}\left( \frac{1}{|\mathcal{M}_t(j)|} \left\| \sum_{i\in\mathcal{M}_{t+1}^{b_t}(j)} (x(i) - c_t(j)) \right\|^2 \right) \right)$$

Finally, each of the two expectation terms can be approximated by $\hat{\sigma}_S^2(j)$. Approximating the first term by $\hat{\sigma}_S^2(j)$, may be an underestimation as the summation is over data which was not used to obtain $c_t(j)$, whereas $\hat{\sigma}_S^2(j)$ is obtained using data used by $c_t(j)$. Using this estimation we get,

$$\approx \frac{1}{2|\mathcal{M}_t(j)|} \hat{\sigma}_S^2(j),$$

$$= \frac{1}{2|M_t(j)|^2} \sum_{i\in\mathcal{M}_t(j)} \|x(i) - c_t(t)\|^2,$$

$$= \frac{1}{2} \hat{\sigma}_C^2(j).$$

The final equality following from the definition of $\hat{\sigma}_C^2(j)$.

## SM-C   Time-energy curves with various doubling thresholds

Figures 4 and 5 show the full time-energy curves for various values of the doubling threshold $\rho$, for the cases where bounds are used and deactivated respectively.

## SM-D   On algorithms intermediate to `mbatch` and `nmbatch`

The primary argument presented in this paper for removing old assignments is to prevent a biased use of samples in `nmbatch`. However, a second reason for removing old assignments is that they can contaminate centroids if left unremoved. This second reason in favour of removing old assignments is also applicable to `mbatch`, and so it is interesting to see if `mbatch` can be improved by removing old assignments, without the inclusion of triangle inequality based bounds. We call this algorithm `mbatch.remove`. In addition, it is interesting to consider the performance of `nmbatch` without bound testing. We here call `nmbatch` without bound testing `nmbatch.deact`.

In Figure 6 we see that `mbatch` is indeed improved by removing old assignments: `mbatch.remove` outperforms `mbatch`, especially at later iterations. The algorithm `nmbatch.deact` does not perform

Figure 4: Time-energy curves for `nmbatch` with various $\rho$. The dotted vertical lines correspond to the time slices presented in Figure 2. We see that large $\rho$ works best, with very little difference between $\rho = 10^2$ and $\rho = 10^3$.

as well as `nmbatch`, as expected, however it is comparable to `mbatch.remove`, if not slightly better. There is no algorithmic reason why `nmbatch.deact` should be better than `mbatch.remove`, as nesting was proposed purely as way to better harness bounds. One possible explanation for the good performance of `nmbatch.deact` is better memory usage: when samples are reused there are fewer cache memory misses.

## SM-E   Premature fine-tuning, one more time please

The loss function being minimised changes when the mini-batch grows. With $b_t$ samples, it is

$$E(\mathcal{C}) = \frac{1}{b_t} \sum_{i=1}^{b_t} \arg\min_{j \in \{1,...,k\}} \|x(i) - c(j)\|^2,$$

and then with $2b_t$ it is

$$E(\mathcal{C}) \frac{1}{2b_t} \sum_{i=1}^{2b_t} \arg\min_{j \in \{1,...,k\}} \|x(i) - c(j)\|^2.$$

Minima of these two loss functions are different, although as $b_t$ gets large they approach each each. Premature fine-tuning refers to putting a large amount of effort into getting very close to a minimum with $b_t$ samples, when we know that as soon as we switch to $2b_t$ samples the minimum will move, undoing our effort to get very close to a mimumum.

Coffee break definition: It's like a glazed cherry without a cake, that finishing touch which is useless until the main project is complete. Donald Knuth once wrote that *premature optimization is the*

Figure 5: Time-energy curves for `nmbatch` with bounds disabled. The dotted vertical lines correspond to the time slices presented in Figure 2, that is $t = 2s$ and $t = 10s$. We see that with bounds disabled, $\rho = 10^1$ in general outperforms $\rho \in \{10^2, 10^3\}$, providing empirical support for the proposed doubling scheme.

*root of all evil*, where optimisations to code performed too early on in a project become useless as software evolves. This is roughly what we're talking about.

Figure 6: Performace of algorithms intermediate to `nmbatch` and `mbatch`. The intermediate algorithms are : `nmbatch.deact`, which is `nmbatch` with the bound test deactivated, and `mbatch.remove`, which is `mbatch` with the removal of old assignments. `nmbatch` and `nmbatch.deact` are with $\rho = 100$ as usual. We observe that, as expected, deactivation of the bound test results in a significant slow-down of `nmbatch`. We also observe that the removal of old assignments significantly improves `mbatch`, especially at later iterations.