[Reviews · NeurIPS 2016]

Reviewer 1

Summary

The authors provide an acceleration technique for k-means which combines mini-batches with triangle quality exclusion for closest prototype estimation. What is challenging in the approach is to choose nested minibatches (otherwise no expected speedup using triangle inequality) and its proper handling of duplicates without bias, and how to extend the minibatches in an optimum way balancing convergence and novelty. For the latter, some nice computations of the expected change are done. Given the relevance of k-means, this contribution seems interesting as concerns applications.It is interesting as concerns algorithm design.

Qualitative Assessment

I think the paper is moslty nice, well written. No stunning new ideas but a very good combination of interesting approaches into one really combined one. It is not so clear to me how the sensitivity of k-means to initialization is dealt with (what happpens in your data for extreme trend? What if prototypes are trapped, I am missing illustrative extremal experiments in this regard) and how the number of clusters is adjusted (which would be reasonable to do if minibatches are considered anyway).

Confidence in this Review

2-Confident (read it all; understood it all reasonably well)


Reviewer 2

Summary

This paper proposes a simple yet efficient improvement of k-means clustering.

Qualitative Assessment

This paper is overall clearly written and easy to read, and there is no fatal flaws. Although the proposed approach is a simple additional trick of mini-batch clustering, it has experimentally shown to be efficient and orders of faster than state-of-the-art k-means algorithms. Since k-means is widely used in many applications, this paper has an impact and can appeal to a large number of people. It would be more interesting if the update condition (L.36 of Algorithm 5) is theoretically analyzed. For example, how is the error bound of the proposed method compared to the mbatch algorithm? Minor comments: - L.135: "tTherefore" -> "Therefore" - L.312: "sparse sparse" -> "sparse"

Confidence in this Review

2-Confident (read it all; understood it all reasonably well)


Reviewer 3

Summary

The paper presents an accelerated implementation of mini-batch k-means algorithm, with supported good empirical performance.

Qualitative Assessment

Technical quality: It seems the nested-batch method is likely to introduce overhead by keeping all previously sampled points in memory, especially since mini-batch k-means is usually run for many iterations? And the computational cost of checking whether a point is already sampled grows as the number of iteration grows as well. How did this not seem to have an effect in your experiments, as comparing to the original mini-batch algorithm? The experiments in Fig. 2 may be a little misguiding: it shows that nested-mini-batch achieves same level of k-means cost faster than the other compared methods; however, this may only mean that it plateaued faster. As time increases, it's possible that the other algorithms will achieve a lower k-means cost eventually (they reach a plateau with a lower k-means cost). The authors do not discuss whether the experiments reveal this feature of nested-mini-batch k-means. Novelty: The paper mostly adapts the idea of distance bounding using triangle inequality (Elkan) to the setting of mini-batch k-means. Potential impact: This paper may have marginal contribution to the existing mini-batch method. Clarity and presentation: The paper is well written and the ideas are clearly presented.

Confidence in this Review

2-Confident (read it all; understood it all reasonably well)


Reviewer 4

Summary

This paper introduces a variant of mini-batch K-means, originally described by Sculley. This variant is based on using (nested) mini-batches and keeping the lower bounds of the samples. By using the triangle inequality, this algorithm reduces the number distances to be computed, and speeds up the original mini-batch algorithm.

Qualitative Assessment

The algorithm is exceptionally well explained and reasoned. Authors introduce the problem correctly, including the concrete algorithms that motivated this new k-means based on nested mini-batches. The reasoning is very easy to follow. Authors already state the "drawbracks" of the algorithm, but this reviewer would like to remark the difficulty of fine-tuning the size of the mini-batches. Experiments are relevant, as they compare the proposed algorithm against others in the literature, over different datasets. A small note on the comparison based on execution time (Appendix A): algorithms should be written in the same language (at least) to have a fair comparison.

Confidence in this Review

2-Confident (read it all; understood it all reasonably well)


Reviewer 5

Summary

The paper presents a new method to accelerate k-means clustering algorithm. The main contributions of this paper are 1. Bounding method to limit the calculations involved in the assignment step 2. Nested batches to avoid duplicitous contribution in update step. The method trades space complexity for acceleration. Method is compared with standard k-means implementations which address similar issues. Results indicate significant improvement in the running time.

Qualitative Assessment

1. The need for a faster algorithm is not correctly motivated 2. Usage of bounding and nested batches are clever ideas 3. However, the definite increase of space complexity (all data samples need to have a bounds vector) of this algorithm over other implementations is not discussed in the paper 4. Theoretical bounds of the running times could help 5. The repeatability of clusters is big concern in k-means, which is not discussed or addressed in this paper 6. The idea of using training and testing data is not clear in the results section 7. The merit of using nested batches is not revealed via the results. May be the authors can add results for bounding + mbatch and compare it with bounding + nested batch. 8. Typos in line 134, 190

Confidence in this Review

2-Confident (read it all; understood it all reasonably well)


Reviewer 6

Summary

The paper provide a fast method of classifying data by using nested mini-batches and through the incorporation of triangular inequality based distance bounds into the algorithm.

Qualitative Assessment

The paper propose an improved k-means clustering method by combining two previously existing algorithms and solving the two potential difficulties. The paper is well written and well organized and the claim are supported by well-explained simulation results. However, there are so many typos and grammatical errors in the paper that need to be corrected before the paper can be published. Here are some of these errors: 1- line 65: initialised-> initialized 2- line 78: note than-> note that 3- line 105: to performing-> to perform 4- line 134: tTherefore-> Therefore 5- line 312: sparse sparse-> sparse -also note that algorithms 2 and 3 are mentioned in the paper without referring to them in the text. -It would be great if the authors can explain why their algorithm doesn't perform very well on sparse datasets.

Confidence in this Review

1-Less confident (might not have understood significant parts)